# A Multi-Echelon Network Design in a Dual-Channel Reverse Supply Chain Considering Consumer Preference

**DOI:** 10.3390/ijerph18094760

**Published:** 2021-04-29

**Authors:** Peng Li, Di Wu

**Affiliations:** Faculty of Economics and Management, Xi’an University of Technology, Xi’an 710048, China; lipenggn@xaut.edu.cn

**Keywords:** dual-channel reverse supply chain (DRSC), network design, consumer preference, ε-constraint method

## Abstract

The rapid development of e-commerce technologies has encouraged collection centers to adopt online recycling channels in addition to their existing traditional (offline) recycling channels, such the idea of coexisting traditional and online recycling channels evolved a new concept of a dual-channel reverse supply chain (DRSC). The adoption of DRSC will make the system lose stability and fall into the trap of complexity. Further the consumer-related factors, such as consumer preference, service level, have also severely affected the system efficiency of DRSC. Therefore, it is necessary to help DRSCs to design their networks for maintaining competitiveness and profitability. This paper focuses on the issues of quantitative modelling for the network design of a general multi-echelon, dual-objective DRSC system. By incorporating consumer preference for the online recycling channel into the system, we investigate a mixed integer linear programming (MILP) model to design the DRSC network with uncertainty and the model is solved using the ε-constraint method to derive optimal Pareto solutions. Numerical results show that there exist positive correlations between consumer preference and total collective quantity, online recycling price and the system profits. The proposed model and solution method could assist recyclers in pricing and service decisions to achieve a balance solution for economic and environmental sustainability.

## 1. Introduction

With the wave of industrialization sweeping the world, consumption of a wide variety of electronic products has grown globally, resulting in the mass of electronic waste (e-waste) is becoming the fastest growing waste flow in the world. It is reported that global e-waste generation was estimated to be 41.8 million tons in 2014 and increased sharply to 60 million tons by 2018, which brought a huge challenge in recovery and disposal [1]. However, only an estimated 20% of global e-waste is fully recycled and safely disposed of. Due it being composed of numerous toxic elements, e-waste can be regarded as hazardous waste; when it is disposed of without care, a great threat is posed to both the environment and human health, and further pressure put on the environment in the form of global warming, resource depletion and the extinction of species (UNEP, 2016). Thus, sustainable development, addressing economic benefits, and environment impacts, receives growing attention in the recycling industry of e-waste. Facing this grim situation, many firms have started to adopt the practice of recovering potential value from e-waste and integrated recovery activities and environmental strategies into their recovery processes, such as HP, IBM, Apple, Kodak, and Lenovo [2]. Likewise, governments have introduced legislation to reduce the serious environmental pollution and this affects how firms take responsibility for the proper handling of their e-waste. All these issues mean that researchers pay great attention to the reverse supply chain (RSC), which can be defined as the reverse flow of a series of facilities including product recovery, transportation, sorting, dismantling and remanufacturing for the purpose of taking back the e-waste and regaining product value in the supply chain. The research on RSC has evolved over the years and has concluded on different aspects. To deal with the uncertainties associated with RSC operations, one of the core challenge problems at the strategic level is to build an advanced stochastic model to generate a network design that will perform well under different scenarios, thereby improving the overall supply performance and profitability [3,4,5]. The network design of RSC has traditionally focused on network, logistics processes, and managing efficient recovery. However, with an ever-increasing concern for environmental sustainability, it is well recognized that firms are not only aiming for an efficient network design, but also seeking to achieve a transformation to a green and sustainable supply chain. Therefore, this paper aims to propose a new generic stochastic network design model which considers key uncertain parameters that are recurrent in recycling industrial of e-waste with the challenges of RSC operations, and the sustainable goals including both economic and environmental factors are also considered [6,7,8].

On the other hand, one of the main reasons for the low efficiency and high cost of e-waste recycling is that there are only a few recycling channels accessible to consumers. With the development of information technology, one possible way to provide accessible services to individuals is to combine the Internet with traditional industries to promote communication between recycling practitioners and consumers. Under such a circumstance, “Internet + recycling”, a new business mode has come into being; this mode enables consumers and recycling practitioners to schedule on-site collections and transactions through various online platforms. Since this model has provided convenience and availability, for any one can register an account on online platforms and then strike a bargain, and also break through the constraints of time and space of traditional recycling channels, it is becoming more and more popular. For example, Changhong Green Group Company limited and Shanghai Xin Jinqiao Environmental Protection Company Limited, who earlier collected e-waste through traditional recycling channels, now began to collect through online recycling channels. In addition, GEM, one of the China’s largest recycling enterprises, has vigorously built online recycling channels through cooperation with the online platform Loving Recycling (refer to www.aihuishou.com, accessed on 10 December 2014). As such the idea of coexisting traditional recycling channels and online recycling channels successfully evolved a new concept of the dual-channel reverse supply chain (DRSC), which refers to the e-waste which is collected at a collection center through traditional recycling channels and online recycling channels together. When a DRSC is adopted, the supply chain management (SCM) becomes more complex but is also more profitable for the supply chain [9,10].

However, managing a DRSC has its own challenges, mainly the conflict and competition between online recycling channels and the traditional recycling channels; this will lead to severe market cannibalization, and the main reason is due to its consumer-driven characteristics. In a DRSC, consumers are shown to have a preference to choose online or traditional recycling channels to dispose of their e-waste, and recyclers of different channels are willing to utilize some strategies to attract more consumers to choose their channels and acquire more market share, such as recycling price, channel convenience, service level, etc. In this process, the consumer is the sender of the transaction, and the degree of consumer satisfaction determines whether the transaction can be successfully completed. Due to this, consumer preference has a significant influence on the performance of DRSC [11]. The existing literature explains the key effects of consumer preference in lots of supply chain modes, but rarely involves DRSC. Therefore, how to comprehensively consider traditional and online recycling channels in DRSC and focus on the impact of consumer preference are issues that urgently need to be resolved. Further, the development of the Internet has provided an attractive alternative for consumers to dispose of their e-waste, and the opening of online recycling channels can also contribute to a large demand of e-waste. In this sense, how to the optimize recycling service level has become an effective way to increase the market share of e-waste. This is not only reflected in the design, operation and consumer service response of its website, but also reflected in its large number of offline network design to facilitate potential consumer consultation and on-site delivery.

In the literature, study of network design of RSC has gained great attention of both academic research and industrial practitioners. For more comprehensive and detailed reviews on these works one can refer to the authors of [12,13,14]. To deal with the uncertainty related to RSC operations, many researchers studied advanced stochastic models under different scenarios. Diabat et al. [15] developed a mixed integer non-linear programming (MINLP) model for a multi-echelon RSC network for product returns. Soleimani and Govindan [16] developed a risk-averse two-stage stochastic programming model for an RSC network by taking the expectation of random variables into consideration. The numerical results proved the capabilities and acceptability of the proposed model and the effects of risk parameters in the model behavior. Roghanian and Pazhoheshfar [17] proposed a probabilistic mixed integer linear programming (MILP) model for a multi-product, multi-stage RSC network with the degree of uncertainty in terms of capacities, demands and return quantity. Zhou and Zhou [18] studied a nonlinear integer programming model for an office paper reverse logistics network to determine the locations and number of recycling stations and plants while minimizing the total cost. Ene and Ozturk [19] considered a network for reverse flows of the end-of-life vehicles’ recovery process. The main objective of the model is to maximize revenue and minimize pollution in end-of-life product operations. Ayvaz et al. [20] proposed a generic multi-echelon, multi-product and capacity-constrained stochastic RSC network design model under return quantity, sorting ratio and transportation cost uncertainties. Numerical examples showed that the proposed model can provide acceptable solutions to make efficient decisions. Fattahi and Govindan [21] considered design and planning for an integrated forward/reverse logistics network over a planning horizon with multiple tactical periods. They applied a Latin Hypercube Sampling method and backward scenario reduction technique to deal with demand and potential return uncertainty. Yu and Solvang [22] presented a stochastic model for designing and planning a model in which they considered a generic multi-source, multi-echelon, capacitated and sustainable RSC network under uncertainty. Srinivasan and Khan [23] presented a manufacturing/re-manufacturing facility location and allocation model for a multi-stage, multi-product capacitated closed-loop supply chain (CLSC) network. To handle the uncertainty with demand and return, a scenario-based mixed integer linear programming was developed and implemented in a cartridge manufacturing industry.

Apart from the uncertainty, a large number of studies pay attention to the sustainable supply chain network design due to the growing environmental impact, legislation and corporate social responsibility [24,25,26,27]. Jin et al. [28] demonstrated that the sustainability requirement might reshape the optimal structure of a supply chain. Utne [29] pointed out that sustainable cost evaluation not only includes economics profits or costs but also considers environmental and social impacts. Due to a great performance in environmental benefits, CO_2_ emission control has been widely employed to the well-designed sustainable RSC network. Hong et al. [30] investigate the impact of emission constraints on a sustainable supply configuration with guaranteed service time. Zohal and Soleimani [31] presented CLSC systems involving four stages of forward flow and three stages of reverse flow, and a multi-objective model was developed with a significant consideration of CO_2_ emission. Yu and Solvang [32] constructed a multi-product, multi-echelon stochastic programming model considering carbon constraint under uncertainty and developed a multi-criteria scenario-based risk-averse solution method to obtain the optimal solutions. Haddadsisakht and Ryan [33] presented a three-stage hybrid robust/stochastic model for CLSC network design problem with carbon tax rate uncertainty, the results showed that the ability of adjusting transportation mode capacities to the tax rate can provide valuable benefit. Zarbakhshniz et al. [34] presented a multi-stage, multi-product and multi-objective model for a CLSC network for the purpose of minimizing the economic costs and CO_2_ emissions; they also developed an ε-constraint method to obtain a set of Pareto solutions. Wang et al. [35] developed a green urban CLSC network to minimize the CO_2_ emissions and the overall operational cost and a case study is conducted to validate the feasibility and practicality of the proposed model.

Moreover, we also discussed the interaction between different stakeholders, due to its particular importance in the context of RSC network design. Evidence had shown that the core firm needs to maintain a healthy relationship with its stakeholders related to RSC and respond positive to their demands, in order to achieve competitive advantages and corporate sustainability [36,37]. De Figueiredo and Mayerle [38] analyzed the individual behavior of recyclers and collection centers, they developed a two-level model to optimize the total system cost by given the behavior of collection centers. Litvinchev et al. [39] proposed a pricing strategy model with multiple periods and stochastic demand, they pointed out that the recovery price and quantity of returned products can be set by the unit cost saving and competitor’s price. Rezapour et al. [40] presented a bi-level model for the CLSC network design with price-dependent market demand, and considered the competition between two chains producing commodities in a same market. Thus, in this study, we use the stakeholder theory as a decision-making tool for firms in order to gain competitive advantages.

Different to the booming development in recycling industry, there is limited research that focuses on DRSC; only a few articles showed up in the research results. Earlier studies demonstrated that the application of Internet technology in the product recovery could effectively collect and analyze the information needed to be recycled, and strengthen the management of product life cycle [41]. Some researchers discussed the pricing and strategic planning of DRSC with the consideration of consumer preference, the main results of their work showed that the DRSC strategy always outperform than the traditional offline channel, and the consumer preference plays an important role in the coordination mechanism of DRSC [42,43]. Giri et al. [44] studied the pricing strategy of DRSC when different members occupied the dominant position, such as manufacturers, retailers and third-party, they found that the optimal revenue can be derived when the retailer dominated the whole system. Moreover, a revenue-sharing mechanism of DRSC were developed by considering the relationship between the recovery rate and the revenue sharing ratio [45]. Taleizadeh et al. [46] investigated the pricing strategies of two types of CLSC with a dual-channel, they developed Stackelberg game models to explore optimal solutions, such as prices, quality levels and collective efforts.

To sum up, scholars have conducted in-depth research for RSC network design optimization problems, especially those that integrates environmental issues, multiple uncertainties and capacity constraints. All these assorted modelling frameworks, solution methodologies and stochastic models in previous studies can provide great help and support to this research. Through analysis of the model and the main results, we have drawn some meaningful points in this research. First of all, although there has only been little research on DRSC, it is still the mainstream research background on supply chains in academia; this is mainly because the dual-channel supply chain with forward supply chains (FSC) and reverse supply chains with multiple offline channels has been become a research trend. Moreover, the research of DRSC, which integrates the Internet, recycling, policy, environment, social and other academic hotspots, will surely become one of the most popular research issues in the future. It is also because of this that the exploration of such problems at this stage is of great significance. Secondly, the majority of the existing models on DRSC are mainly focused on pricing decisions and channel coordination [42,43,44,45,46]; there is a lack of quantitative network design models that represent advanced applications in recycling industries; the DRSC that considers both traditional and online recycling channels remains scarce. Thirdly, some studies considered consumer preference as a key consumer-related factor to the system performance of supply chain and showed that it will cause the system to lose stability and fall into the trap of complexity [47]. Most previous studies have studied its impacts on the forward supply chain or RSC, only the work of [43,45] considered contract design and strategy selection for the DRSC. To fill this gap, this work contributes to the existing literature by formulating a stochastic mathematical model for the network design of a general DRSC system and, through the numerical analysis, we have explored the impact of consumer preference on the channel selection, pricing, and revenue of recycling practitioners in DRSC and its deeper management significance.

To the best of our knowledge, this paper is the one of the few studies on DRSC based on online recycling channels that simultaneously considers environmental issues and the impact of consumer preference in designing the DRSC network. Specifically, we design the network of a DRSC system that integrates traditional and online recycling channels. Through the construction and solution of a mixed integer linear programming (MILP) model, we analyze the examples and supply chain systems change with the increase in consumer preference and service cost coefficient. The goal is to minimize the total costs while reducing the carbon emissions by strategically locating participants within the DRSC network (such as recyclers, disposal centers, remanufacturing centers, etc.). To efficiently solve the model, an ε-constraint method is proposed, and the effectiveness of the solution method is validated by three sizes of test problems: small, medium and large. Briefly, the main contributions of this work are as follows:a new MILP model for a stochastic DRSC network design considering consumer preference is proposed;dual objectives including the total costs and carbon emissions are considered;the uncertainties in demand and collective quantity are considered;an ε-constraint algorithm to cope with the large-scale RSC network design problem is developed.

The remainder of this paper is organized as follows: Section 2.1 gives the main assumtions for the model and Section 2.2 describes the studied problem in detail and introduces the process of formulating the bi-objective stochastic MILP model. Section 2.3 presents an ε-constrained method that improves the efficiency of the problem-solving. In Section 3, the examples of the model are tested and validated. Finally, the conclusions and future research are given in Section 5.

## 2. Materials and Methods

### 2.1. Assumptions

The following assumptions are made to develop the proposed model:

**Assumption** **1.***In order to simplify the study, according to the research in* [42,43,48,49]*, we assume that all the e-waste products are of the same type, the same degree of loss and recovery conversion. Online recycling companies can recycle almost all kinds of e-waste, such as mobile phones, notebook computers, digital cameras and other types, and there is no e-waste with absolutely the same degree of loss in reality. However, since none of the above is the focus of this research and will make the model too complicated, in this study we only consider a batch of e-waste of the same type, brand, and degree of loss that have been sorted by collection centers.*

**Assumption** **2.**
*We assume that the online recycling platform is established by third-party recyclers (TPR), which is currently the main mode of constructing such platforms in China, accounting for about 80% of the total number. Some manufacturers, while recycling e-waste through existing channels, such as offline stores and official websites, also entrust online recycling platforms to recycle. For example, in China, Meizu Inc. recycle used phones through its self-built “mCycle” recycling project and cooperate with “ihuigo network” (http://www.ihuigo.com, accessed on 3 August 2017) for recycling; Huawei Inc. carries out part-exchange activities and recycles in cooperation with “huishoubao network” (http://www.huishoubao.com, accessed on 17 August 2016). Such a new mode can reduce the cost of self-built, and outsourcing services can also improve service quality.*


**Assumption** **3.***In the existing research describing online recycling channels* [44]*, they always assume that collection centers can collect e-waste directly from consumers but without going through any self-operated offline stores. That is, once consumers reach an agreement with online recyclers for the recycling price through Internet, they will send the e-waste to collection centers by express delivery or door-to-door. However, according to the research in* [50]*, in reality we found that almost all online recycling channels have built their own self-operated stations to support online recycling transactions. For example, Loving Recycling collects all types of used mobile phones, electronic instruments, cameras, etc. through the Internet; they also build offline stores, such as the eighty-eight offline stores built in Shanghai and fifty-eight in Beijing. Therefore, in this study, self-operated stores built by online recyclers will be considered in the DRSC network studied, and the e-waste collected by online recycling channels will be transferred to collection centers after initial sorting.*


**Assumption** **4.***Based on a large number of studies such as* [48,51]*, we assume that the collective quantity of online and traditional recycling channels is linearly affected by recycling prices and service level. In addition, we believe that consumer preference for the online recycling channel θ (0 < θ < 1) will affect the proportion of the basic quantity of the recycling market in different channels, and this proportion is not affected by recycling prices and service level. Due to the competition between the two channels in the recycling market, the recycling price of one party will affect the collective quantity of the other party. Therefore, if we define p_t_ and p_e_ as the recycling price of unit e-waste to consumers in traditional and online recycling channels, respectively. Then, the collection quantity of traditional and online recycling channels is:*

*D_r_* = (1 − *θ*)*α* + *δ*_1_*p_t_* − *δ*_2_*p_e_* − *η*_2_*s*_l_(1)

*D_s_* = *θα* + *δ*_1_*p_e_* − *δ*_2_*p_t_* + *η*_1_*s*_l_(2)


*where a is the basic market share to the direct channel or in, other words, the web-product compatibility quantity of the recycling market; δ_1_ is the coefficient of collective quantity affected by the recycling price of own channels, and δ_2_ is the coefficient of collective quantity affected by the recycling price of competing channels; η_1_ is the coefficient of collective quantity affected by the service level of own channels, and η_2_ is the coefficient of collective quantity affected by the service level of competing channels; s_l_ is service level of online recycling channels.*


**Assumption** **5.***Similar to the research in* [52]*, we also assume that traditional local recyclers are mostly small-scale recyclers, and always located near consumers. For most types of e-waste, as the traditional recycling mode usually obtains bulk transportation; the logistics cost shared by each used product is very low, which can be ignored. Therefore, we assume that the location of the local recycler is the consumer area, and the distance from consumer to the location recycler is ignored.*

**Assumption** **6.***In this study, we also choose to ignore the service level of traditional recycling channels when building the model* [53]*. Although traditional recycling companies provide consumers with consulting and pickup services when dealing with consumers, their service level are too low compared to online recycling channels. In addition, online recycling companies have fixed standards and procedures for service provision and have used them as an important means to enhance channel competitiveness, which is also not available in traditional recycling companies.*

### 2.2. Model Formulation

In this study we consider a general network for the DRSC that integrates traditional and online recycling channels to optimize the operations of product recovery and remanufacturing. This is designed as a multi-echelon reverse logistics network with five members—local recyclers of traditional recycling channels, third-party recyclers of online channels (TPR), collection centers, remanufacturing centers and disposal centers. It has a dual-channel recycling system by which the e-waste products are collected from consumers through traditional local recyclers or TPR to collection centers. After proper inspection and grading at collection centers, the e-waste will be classified and distributed via different reverse channels. Most e-waste which can be repaired will be sent to remanufacturing centers for remanufacturing and be resent back to consumers, whereas a small number of scrapped e-waste will be transported to disposal centers for refuse disposal, such as landfill or incinerators. Therefore, the DRSC network proposed in this paper can be described as shown in Figure 1.

The objectives of the proposed model include two parts: one is to minimize the total operating costs of the entire network (*OF*_1_), which consist of fixed establishing costs of the facilities (*f*_1_) and transporting costs for delivering the e-waste (*f*_2_); another is to measure the impact of environmental factors on the network. This expression aims to minimize the total amount of CO_2_ emissions generated from transport flow and facility operations (*OF*_2_), which consist of the amount of CO_2_ emissions for establishing the facilities (*f*_3_), for handling the e-waste of each facility (*f*_4_), and for transporting the e-waste between facilities (*f*_5_). 

We first introduce the indices, parameters and decision variables of the proposed model in Table 1, respectively.

Next, we formulate the equations of five subitems involved in two objectives as follows: the fixed establishing cost of facilities can be formulated by
(3)$$f_{1}=\underset{j\in J}{\sum }FT_{j}w_{j}+\underset{k\in K}{\sum }FC_{k}x_{k}+\underset{l\in L}{\sum }FR_{l}y_{l}+\underset{m\in M}{\sum }FD_{m}z_{m}$$the transporting cost for delivering e-waste is formulated by
(4)$$f_{2}=\underset{s\in S}{\sum }P_{s}\;\left(\underset{i\in I}{\sum }\underset{j\in J}{\sum }PT_{ij}a_{ij}^{s}+\underset{i\in I}{\sum }\underset{k\in K}{\sum }PC_{ik}\gamma_{i}^{s}+\underset{j\in J}{\sum }\underset{k\in K}{\sum }PL_{jk}\beta_{jk}^{s}+\;\underset{l\in L}{\sum }\underset{k\in K}{\sum }PR_{kl}\delta_{kl}^{s}+\underset{m\in M}{\sum }\underset{k\in K}{\sum }PD_{km}\eta_{km}^{s}+\underset{i\in I}{\sum }\underset{l\in L}{\sum }PU_{li}\theta_{li}^{s}\;\right)$$the total amount of CO_2_ emissions for establishing facilities is calculated by
(5)$$f_{3}=\underset{j\in J}{\sum }ET_{j}w_{j}+\underset{k\in K}{\sum }EC_{k}x_{k}+\underset{l\in L}{\sum }ER_{l}y_{l}+\underset{m\in M}{\sum }ED_{m}z_{m}$$the total amount of CO_2_ emissions for handling e-waste is formulated by
(6)$$f_{4}=\underset{i\in I}{\sum }\underset{j\in J}{\sum }HT_{j}a_{ij}^{s}+\underset{j\in J}{\sum }\underset{k\in K}{\sum }HC_{k}\beta_{jk}^{s}+\;\underset{i\in I}{\sum }\underset{k\in K}{\sum }HC_{k}\gamma_{ik}^{s}+\underset{l\in L}{\sum }\underset{k\in K}{\sum }HR_{l}\delta_{kl}^{s}+\underset{m\in M}{\sum }\underset{k\in K}{\sum }HD_{m}\eta_{km}^{s})$$the total amount of CO_2_ emissions for transporting e-waste is calculated by
(7)$$f_{5}=t\rho \underset{s\in S}{\sum }P_{s}\;\left(\underset{i\in I}{\sum }\underset{j\in J}{\sum }DT_{ij}a_{ij}^{s}+\underset{j\in J}{\sum }\underset{k\in K}{\sum }DC_{jk}\beta_{jk}^{s}+\underset{i\in I}{\sum }\underset{k\in K}{\sum }DL_{ik}\gamma_{ik}^{s}+\underset{l\in L}{\sum }\underset{k\in K}{\sum }DR_{kl}\delta_{kl}^{s}+\underset{m\in M}{\sum }\underset{k\in K}{\sum }DD_{km}\eta_{km}^{s}+\underset{i\in I}{\sum }\underset{l\in L}{\sum }DU_{li}\theta_{li}^{s}\;\right)$$

Therefore, the mathematical representation of the model is presented as follows:*MinOF*_1_ = *f*_1_ + *f*_2_(8)
*MinOF*_2_ = *f*_3_ + *f*_4_ + *f*_5_(9)

The constraints of the model are formulated in Equations (10)–(25).
(10)$$\underset{i\in I}{\sum }\underset{j\in J}{\sum }a_{ij}^{s}=\;D_{s},\;\forall s\in S$$
(11)$$\underset{i\in I}{\sum }\underset{k\in K}{\sum }\gamma_{ik}^{s}=\;D_{r},\;\forall s\in S$$
(12)$$\underset{i\in I}{\sum }a_{ij}^{s}=\;\underset{k\in K}{\sum }\beta_{jk}^{s},\;\forall j\in J,\;\forall s\in S$$
(13)$$\left(1-\beta_{1}\right)(\underset{i\in I}{\sum }\gamma_{ik}^{s}+\underset{j\in J}{\sum }\beta_{jk}^{s})=\;\underset{m\in M}{\sum }\eta_{km}^{s},\;\forall k\in K,\;\forall s\in S$$
(14)$$\;\beta_{1}(\underset{i\in I}{\sum }\gamma_{ik}^{s}+\underset{j\in J}{\sum }\beta_{jk}^{s})=\;\underset{l\in L}{\sum }\delta_{kl}^{s},\;\forall k\in K,\;\forall s\in S$$
(15)$$\beta_{2}\underset{k\in K}{\sum }\delta_{kl}^{s}=\underset{i\in I}{\sum }\theta_{li}^{s},\;\forall l\in L,\;\forall s\in S$$
(16)$$\underset{s\in S}{\sum }\underset{i\in I}{\sum }a_{ij}^{s}\leq \;CT_{j}w_{j},\;\forall j\in J$$
(17)$$\underset{s\in S}{\sum }\underset{i\in I}{\sum }\gamma_{ik}^{s}+\underset{s\in S}{\sum }\underset{j\in J}{\sum }\beta_{jk}^{s}\leq \;CC_{k}x_{k},\;\forall k\in K$$
(18)$$\;\beta_{1}\underset{s\in S}{\sum }\underset{k\in K}{\sum }\delta_{lk}^{s}\;\leq CR_{l}y_{l},\;\forall l\in L$$
(19)$$\left(1-\beta_{1}\right)\underset{s\in S}{\sum }\underset{k\in K}{\sum }\eta_{km}^{s}\;\leq CD_{m}z_{m},\;\forall m\in M$$
(20)$$\underset{j\in J}{\sum }w_{j}\geq 1$$
(21)$$\underset{k\in K}{\sum }x_{k}\;\geq 1$$
(22)$$\underset{l\in L}{\sum }y_{l}\geq 1$$
(23)$$\underset{m\in M}{\sum }z_{m}\geq 1$$
*a^s^_ij_*, *γ^s^_ik_*, *β^s^_jk_*, *δ^s^_lk_*, *η^s^_mk_*, *θ^s^_il_* ≥ 0, ∀*i* ∈ *I*, ∀*j* ∈ *J*, ∀*k* ∈ *K*, ∀*l* ∈ *L*, ∀*m* ∈ *M*(24)
*w_j_*, *x_k_*, *y_l_*, *z_m_* = {0, 1}, ∀*i* ∈ *I*, ∀*j* ∈ *J*, ∀*k* ∈ *K*, ∀*l* ∈ *L*, ∀*m* ∈ *M*(25)

Equations (8) and (9) describe the total costs and the total amount of CO_2_ emissions. Constraints (10), (11) ensure that all e-waste generated from consumers by traditional and online recycling channels is completely gathered. Constraints (12)–(15) are the balance constraints which confirm the uniformity of input flow and output flow at each facility. Constraints (16)–(19) assure that the total flows to and from each facility could not exceed its capacity. Constraints (20)–(23) guarantee that at least one of the potential facilities be selected. Constraints (24), (25) are the positive variables and binary constraints.

### 2.3. Solution Method

In this section, how to optimize the dual-objective function and most complex constraints of DRSC network design problem will be introduced. When considering a multiple objective stochastic problem, the sets of the solutions should be a Pareto frontier that represents the trade-off between multiple objective functions instead of a unique solution. As mentioned in the literature, there are two classes of methods to optimize the presented dual-objective model: the first one is to use meta-heuristics or evolutionary methods to obtain acceptable solutions. However, the quality of the solutions and their optimality are not known. The other type of methods are the exact or heuristic methods used to obtain the Pareto solutions [32,34]. In this study, we perform an ε-constraint method, a well-known exact method for a large number of scenarios considered. Moreover, in view of the impact of pricing and service-making on collective quantity in traditional and online recycling channels, which in turn affects the network design of the DRSC, it is necessary to conduct research on how collective centers can optimize their pricing strategy to coordinate the dual-channel of the DRSC. All these works will be stated in next Section 2.3.1 and Section 2.3.2, respectively.

#### 2.3.1. Pricing Strategy Optimization

This section considers the pricing and service decision issues of the DRSC network under a centralized policy, aiming to derive the collective quantity both in traditional and online recycling channels. Under this policy, all facilities cooperatively decide the recycling price and the service level through online recycling channels. Since there is a single decision maker, the internal transfer price does not play any role. That is, the collection center and TPR will no longer make decisions based on their own benefits, but rather maximize the revenue of the entire network. Therefore, the DRSC network will make decisions on traditional recycling price p_t_, online recycling price p_e_, and online service level s_l_, all these parameters will deeply affect the collective quantity. For convenience, we also define that ∏ is the revenue of the whole supply chain system under a centralized policy, p_0_ is the revenue of collection centers from disposal of unit e-waste, c and c_s_ are the logistics cost and service cost of recycling in online channels, respectively. Moreover, according to Tsay and Agrawal [54], c_s_ can be described as μs_l_^2^/2 where μ represents the coefficient of service cost of online channels. Therefore, the formulation of the whole system revenue (∏) is mainly composed of the revenue of both traditional and online recycling channels, which can be expressed as follows:∏ = (*p*_0_ − *p*_t_)*D_r_* + (*p*_0_− *p*_t_ − *c*) *D_s_* − *c_s_* = (*p*_0_ − *p*_t_)[(1 − *θ*)*α + δ*_1_*p_t_* − *δ*_2_*p_e_* − *η*_2_*s_l_*] + (*p*_0_ − *p_t_* − *c*)[*θα + δ*_1_*p_e_* − *δ*_2_*p_t_ +**η*_1_*s_l_*] − *μs_l_*^2^/2(26)

Next, according to Equation (26), we can solve the first partial derivatives of ∏ with respect to *p_e_* and *p_t_* and *s_l_*, respectively, and the results are given as follows:∂∏/∂*p_e_* = −2*δ*_1_*p_e_ + 2δ*_2_*p_t_* − *η*_1_*s_l_* + (*δ*_1_ − *δ*_2_) *p*_0_ − *θα* − *cδ*_1_(27)
∂∏/∂*p_t_* = −2*δ*_1_*p_t_ + 2δ*_2_*p_e_* + *η*_2_*s_l_* + (*δ*_1_ − *δ*_1_) *p*_0_ − (1− *θ*)*α* + *cδ*_2_(28)
∂∏/∂*s_l_* = −*μs_l_* + *p*_0_(*η*_1_ − *η*_2_) + *p_t_**η*_2_ − *p_e_**η*_1_ − *c**η*_1_(29)

**Property** **1.**
*When η_1_^2^δ_1_ + η_2_^2^δ_1_ − 2η_1_η_2_δ_2_ + 2μ(δ_2_^2^ − δ_1_^2^)*
* < 0, the objective function*
*∏(p_t_, p_e_, s_l_) is concave with p_t_, p_e_ and s_l_.*


The proof of Property 1 can be found in Appendix A.

According to Cachon and Lariviere [55] and Wu et al. [56], the optimal value of *p_t_*, *p_e_* and *s_l_* can be derived by setting Equations (27)–(29) to zero and combining the results as follows:*p_t_* = {*c**η*_1_(*η*_2_*δ*_1_ − *η*_1_*δ*_2_) + *α*[*η*_1_^2^(*θ* − 1) − *η*_1_*η*_2_*θ* + 2*μ*(*δ*_1_ − *θδ*_1_ + *θδ*_2_)] + *p*_0_[2*η*_2_^2^*δ*_1_ − *η*_1_*η*_2_(*δ*_1_ + 3*δ*_2_) + (*δ*_1_ + *δ*_2_)(*η*_1_^2^ − 2*μδ*_1_ + 2*μδ*_2_)]}/2[*η*_1_^2^*δ*_1_ + *η*_2_^2^*δ*_1_ − 2*η*_1_*η*_2_*δ*_2_ + 2*μ*(*δ*_2_^2^ − *δ*_1_^2^)](30)
*p_e_* = {−*c*(*η*_2_^2^*δ*_1_ + 2*η*_1_^2^*δ*_1_ − 3*η*_1_*η*_2_*δ*_2_ − 2*μδ*_1_^2^ + 2*μδ*_2_^2^) + *α*[*η*_1_*η*_2_*θ* − *η*_1_*η*_2_ − *η*_2_^2^*θ* + 2*μ*(*δ*_2_ + *θδ*_1_− *θδ*_2_)] + *p*_0_[*η*_2_^2^(*δ*_1_ + *δ*_2_) −*η*_1_*η*_2_(*δ*_1_ + 3*δ*_2_) + 2(*η*_1_^2^*δ*_1_ − *μδ*_1_^2^ + *μδ*_2_)]}/2[*η*_1_^2^*δ*_1_ + *η*_2_^2^*δ*_1_ − 2*η*_1_*η*_2_*δ*_2_ + 2*μ*(*δ*_2_^2^ − *δ*_1_^2^)](31)
*s_l_* = {*c**η*_1_(*δ*_1_^2^ − *δ*_2_^2^) + *α*[*η*_1_(*θδ*_2_ − *δ*_2_ − *θδ*_1_)+ *η*_2_(*θδ*_2_ − *δ*_1_ − *θδ*_1_)]+ *p_0_*(*δ*_1_^2^ − *δ*_2_^2^)(*η*_2_ − *η*_1_)}/[*η*_1_^2^*δ*_1_ + *η*_2_^2^*δ*_1_ − 2*η*_1_*η*_2_*δ*_2_ + 2*μ*(*δ*_2_^2^ − *δ*_1_^2^)](32)

Moreover, substituting (30)–(32) into (1) and (2), the collective quantity of traditional and online recycling channels can be obtained, respectively.

**Property** **2.**
*Under the centralized policy, the optimal online recycling price p_e_ is negatively correlated with θ, and the optimal service level s_l_ is positively correlated with θ.*


Property 2 (the proof of Property 2 can be found in Appendix A) indicates that when consumer preference for the online recycling channel increases, recyclers need to reduce their online recycling prices while increasing service level to ensure the maximum profits.

#### 2.3.2. ε-Constraint Method

In the field of multi-objective optimization problems, the ε-constraint method is most commonly used to obtain efficient solutions [57,58]. This technique solves the model repetitively in which, for each replication, one objective function is taken as the only objective function while the others are set as constraints using appropriate values [59]. Compared with other multi-objective optimization methods, it has the advantages of efficiently obtaining Pareto solution sets, and no additional parameters or uniform dimensions, etc. The ε-constraint model can be expressed as follows:*Min**f_j_*(*x*)(33)
*s.t. f_j_*(*x*) ≤ *ε_i_*, 1 ≤ *I* ≤ *k*, *i* ≠ *j*(34)
*x* ∈ *X_f_*(35)
where *f_i_*(*x*)(*i* = 1, 2,…, *k*) represents the *i*-th objective of a multi-objective problem with *k* objectives, as the upper bound of the objective function, *ε_i_*, can take different values between its minimum and maximum value to obtain multiple Pareto optimal solutions.

To illustrate the algorithm, we first need to introduce the concept of Pareto dominance. For minimizing the multi-objective function, the Pareto dominance relationship can be defined as: if one feasible solution *X* occupies better than another feasible solution *Y*, then there must be *f*_1_(*X*) ≤ *f*_1_(*Y*) and *f*_2_(*X*) ≤ *f*_2_(*Y*); at least one inequality must be strictly less than the sign. With the concept of Pareto dominance, all undominated points in the feasible solution space of the objective function constitute the Pareto front. On this front, there is no dominant relationship between any two points. In other words, they are “just as good” solutions. This frontier or solution set contains a series of different points that are used by decision makers to make tradeoffs between the values of the objective function.

The main idea of the ε-constraint method is to construct and solve a series of constraint problems by transforming a target into constraints. This series of constraints are linked by decreasing ε values step by step. In order to describe the ε-constraint method, the following three points need to be defined: the Ideal point: set *f*^1^ = (*f*_1_^1^, *f*_2_^1^), where *f*_1_^1^ = min{*f*_1_(*X*)}, *f*_2_^1^ = min{*f*_2_(*X*)}; the Nadir point: set *f^N^* = (*f*_1_*^N^*, *f*_2_*^N^*), where *f*_1_*^N^* = *min*{*f*_1_(*X*); *f*_2_(*X*) = *f*_2_^1^}, *f*_2_*^N^* = *min*{*f*_2_(*X*); *f*_1_(*X*)= *f*_1_^1^}; the Extreme point: set *f^E^* = {(*f*_1_^1^, *f*_2_*^N^*), (*f*_1_*^N^*, *f*_2_^1^)} as the Pareto frontier.

Therefore, the solving procedure of the method can be described as follows:

Step 1: calculate the Ideal points *f*^1^= (*f*_1_^1^, *f*_2_^1^) and Nadir points *f^N^* = (*f*_1_*^N^*, *f*_2_*^N^*);

Step 2: set *F*’ = {(*f*_1_^2^, *f*_2_*^N^*)}, and let *ε* = *f*_2_*^N^* − Δ (Δ = 3);

Step 3: while ε ≥ *f*_2_^1^;

Step 4: solve the ε-constraint problem, where the constraint is *ε* = *f*_2_^*^ − Δ, then the single-objective optimization is to minimize *f*_2_. Solve the single-objective optimization problem to the best, and add the best solution (*f*_1_^*^, *f*_2_^*^) to the set *F*’;

Step 5: By removing the dominant points from the set *F’*, the Pareto frontier *F* is obtained.

## 3. Results

In this section, several computational experiments are established to evaluate the performance of the proposed model and the solution method. Firstly, a sensitivity analysis is conducted to test the impact of key parameters on the network performance of the DRSC. Furthermore, to identify the validity of the proposed dual-objective stochastic mathematical model, three sizes with small, medium and large problems are considered.

### 3.1. Testing the Impact of θ and μ on the Network Performance

Compared to traditional recycling channels, online recycling channels are not only more convenient for consumers and privacy protection, but also more environmentally friendly due to their higher recycling conversion rate and lower pollution emissions. In addition, the increasing popularity of the Internet will always promote consumer preference for the online recycling channel. Based on this, we conduct example analysis aiming at the change of the index of consumer preference for the online recycling channel *θ* and service cost *μ* to the impact on collective quantity of e-waste in both channels and channel member’s profits and explore the causes and future countermeasures based on the analysis of data results and trends.

According to Wu [60] and Xie et al. [61], we give the following parameter settings: *p*_0_ = 1000, *c* = 10, *α* = 500, *μ* = 4, *δ*_1_ = 2, *δ*_2_ = 1, *η*_1_ = 1, *η*_2_ = 2. That is, the basic collective quantity of e-waste is 500 and, for collection centers, the revenue of each unit of e-waste is 1000, and the recovery operation and maintenance cost is *c* = 10. In addition, the practical significance of *δ*_1_ = 2 and *δ*_2_ = 1 means that, when the recycling price of a channel changes by one unit, the impact on the channel’s recycling volume will be two times that of the unit, while the impact of recycling is doubled. The same meaning can also be used to explain *η*_1_ = 1 and *η*_2_ = 2, that is, when the change in the service level of a unit of a channel affects the recovery of dual channels.

We first assume that service cost *μ* is constant and *θ* gradually increases from 0.2 to 0.9. It can be obtained that the pricing, service decisions and profits under centralized policy are shown in Table 2. In addition, the decision results are also drawn in Figure 2 and Figure 3 to show further analysis and practical significance.

From Table 2 and Figure 2, we can demonstrate that the increase in *θ* will lead to an increase in the collective quantity of online recycling channels and a decrease in that of traditional recycling channels. This conclusion conforms to the practice and can be easily reached. Furthermore, it can be found that an increase in this preference will also lead to an increase in the total collective quantity of the system. Consumer preferences for the online recycling channel are only reflected in the distribution of the basic market volume of traditional and online channels and will not directly affect the total collective quantity. Through deep analysis, we can find that it affects online and traditional recycling prices, which in turn changes the total collective quantity. In summary, TPR can increase consumer preferences for the online recycling channel by advertising announcements and policy guidance. This will help increase the total collective quantity, improve the environment, and further promote the recycling industry and the development of recycling enterprise.

In addition, it can be seen from Table 2 that with the increase in *θ*, the total profit of the collection center has shown an increasing trend; this is mainly because more and more consumers will choose online recycling channels to recycle their e-waste, so the increase in total collective quantity will increase the profit of collection centers. Additionally, the promotion of *θ* also made consumers who choose traditional recycling channels switch to online recycling channels. However, for the unit of e-waste, the profits made by online recycling channels are higher than those of traditional recycling channels. It has gradually increased the total profit of the supply chain. In summary, collection centers should actively use advertising, policy guidance and other methods to promote consumer preference for the online recycling channel. This is not only beneficial to the improvement of the profit of itself and the supply chain but also promotes the further exploration of the potential recycling market.

Secondly, from Figure 3, in order to increase the total profit of collection centers, the system should reduce the online recycling price *p_e_*, and improve the traditional recycling price *p_t_* and service level *s_l_*. This is because, as more consumers choose online recycling channels, collection centers can recycle e-waste at a lower recycling price. However, in order to ensure the continuous increase in the collective quantity in online channels, collection centers still need to improve their service level. In traditional recycling channels, in order to retain more consumers, local recyclers have to increase their recycling price. At the same time, for maintaining their own profit level and ensuring the coordination of supply chain, collection centers will also choose to increase the online transfer price for local recyclers. In summary, although collection centers can reduce online recycling price to consumers with the increase in *Θ*, they still need to improve their service level. In addition, in order to help the local recycler maintain a certain level of revenue, collection centers also need to increase their offline transfer price while the local recycler increases the traditional recycling price. As can be seen from the above figure, along with consumer preference for the online recycling channel, the supply chain system should gradually the increase recycling price of traditional recycling channels and the recycling service level of online channels, while reducing the online recycling price to profit optimization.

Next, we will analyze the impact of the service cost coefficient *μ* of recycling service on the decision and profits. Since the effect of *μ* on pricing and service decision is not an upper or lower convex function in the model’s solution result, we need to study it through example analysis. In reality, with the improvement of recycling business processes, and in the case of providing an equivalent recycling service, the service cost coefficient can be reduced by continuously improving service efficiency and streamlining service processes. Based on this, we assume that *θ* = 0.4 remains unchanged, and the service cost coefficient *μ* gradually decreases from nine to three, and the competitional results are given in Table 3, Figure 4 and Figure 5.

From the above results, we can demonstrate that the reduction in *μ* will lead to a decrease in the collective quantity in traditional recycling channels, an increase in that in online recycling channels and the total amount of the supply chain system. This is mainly because a decrease in *μ* can promote collection centers to increase their service level, which in turn can increase online and total collective quantity. At the same time, consumers in traditional recycling channels will also switch to online channels due to their high service level, resulting in a reduction in the collective quantity in traditional channels.

It is worth mentioning that with the reduction in the service cost coefficient *μ*, the optimal recycling price *p_e_* gradually decrease, while service level *s_l_* increases. However, when *μ* reaches below five, the decision of the collection center is particularly sensitive to it. In addition, the recycling price of traditional recycling channel *p_t_* is not affected by *μ*. Although when *μ* decreases, its impact on the decision of recyclers becomes more significant but, in fact, the optimization of decisions does not always exist if all goes well. That is, collection centers can easily reduce it from nine to eight, but it is difficult to reduce it from four to three. Finally, as the service cost coefficient *μ* decreases, collection centers need to reduce their online recycling price or increase the level of recycling services to optimize their profits.

### 3.2. Testing ε-Constraint Algorithm

In this section, three levels of instances with small, medium and large sizes are generated and, for each level, a set of test problems are generated in order to analyze the efficiency and performance of the solution method. More specifically, the setting of test data can be described as follows: (1) the dimensions of test problems are shown in Table 4, in which the number of potential places in each level directly indicates the complexity of the problem; (2) similar to the previous research in [16,20,33], the main input parameters involved in the model are shown in Table 5. (3) Based on the urban population distribution and the demand records of e-waste, the potential position of all facilities is given. By calculating the distance between two facilities by the longitude and latitude coordinates between the two facilities, then the detailed geographic coordinate information about facilities is derived as shown in Table 6, Table 7, Table 8, Table 9, Table 10 and Table 11 (unit: km).

According to the above data, three sets of Pareto solutions can be derived through GAMS software on test problems as shown in Table 12, Table 13 and Table 14, which show the two objective values of the proposed model. Khalili-Damghani et al. [62] pointed out that the solutions on the Pareto frontier in the dual-objective model are a set of non-dominated solutions, and the decision-maker chooses the best solution with respect to their goal and the Pareto solutions.

For more visualization, Figure 6 shows the Pareto frontier output with the dual target for each size of the test problem. If these points are connected into a smooth curve, any point on the curve can be used as the optimal decision plan. From Figure 6, we can demonstrate that increasing the value of the first objective function will cause the value of the other objective function to deteriorate; that is, these two objectives conflict with each other.

Moreover, we also highlight the objective values of the problem in Figure 7, which show the values of costs and CO_2_ emissions with three sets of instances on test problems, respectively.

In Figure 7 it is shown that, as the size of test problems increases and the number of test problems increases, there is no large-scale oscillation in the value of the first objective function; only a small decrease trend occurs in all three sizes of instances. Next, it also appears that the second objective function is increased in all three sizes of instances.

## 4. Discussion

In this subsection, we summarize and discuss the results of example analysis in Section 3.1 and Section 3.2, respectively. First of all, online recycling channels can be used as a lever to force the recyclers to enhance the recycling price of traditional recycling channels, further helping the collection centers increase the total collective quantity while improving the overall profits of the system. In other words, the DRSC benefits both the collection centers and the total system. Thus, with the legislations of environmental protection and corporate social responsibility, the opening of the DRSC strategy can help firms perform their duties and promote their green image. In addition, based on a DRSC system with traditional recycling prices and online recycling prices, we found that consumer preference for online recycling channels plays an important role in the channel selection of collection centers; this can provide guidance for the firm’s decision-making.

Secondly, according to the research, the dual-objective optimization problem needs to explore the Pareto boundary curve, which can provide firms with a combination strategy of the “best solution”. Figure 6 shows the Pareto optimal curve of the proposed model, which demonstrates the interaction between costs and carbon emissions. It can be demonstrated that, if firms want to reduce carbon emissions, they need to increase additional investment funds. From the perspective of environmental impact analysis, the increase in corporate CO_2_ emissions is mainly reflected in the fixed establishment of facilities and transportation flow among facilities.

## 5. Conclusions

This paper investigates the strategic network design of a general multi-echelon, dual-channel reverse supply chain (DRSC) with traditional and online recycling channels. By incorporating consumer preference for the online recycling channel into the system, a detailed quantitative model for the network design optimization is constructed where the collective quantity of each recycling channel relies on the consumer’s willingness. Furthermore, we extensively investigated the pricing strategy to explore the dynamic characteristic of consumer preference for the online recycling channel and service costs on collective quantity, recycling prices and system performance of the DRSC. Based on the above, an ε-constraint method is developed and fitted to the proposed model to obtained the Pareto frontier of the objective functions, which emphasized simultaneously both the economic profits and environmental sustainability. Finally, three types of test problems with small, medium and large sizes are generated and coded in Gams to yield the set of Pareto solutions. The main findings of this study are summarized as follows:

First, the stochastic dual-objective model proposed in this study can effectively solve the network design optimization of the DRSC system. To illustrate the trade-off between CO_2_ reduction and cost control, the estimated Pareto frontier of test problems are derived by the Pareto curves (as shown in Figure 6). Numerical experiments show the effectiveness of the proposed ε-constraint method, and the solution has also achieved significant improvement in solving a large-scale DRSC network.

Second, the consumer preference and unit service cost for the online recycling channel play important roles in the operation of dual-recycling channels. The increase in consumer preference and decrease in unit service cost for the online recycling channel will lead to a decrease in collective quantity of traditional recycling channels, but the collective quantity of online recycling channels and the whole system are still on the rise, which also makes the total profits of the system increase. This is because consumers’ increased preference for the online recycling channel can make more consumers choose to recycle their idle e-waste in the market. In addition, the reduction in unit service cost can prompt companies to improve service level and reduce online recycling price, while still ensuring that the collective quantity increases. Therefore, recyclers should improve the consumer preference for the online recycling channel through advertising, policy advocacy and consumption guidance, and reduce unit service costs by improving service technology and processes, so as to help recyclers improve profits, and promote the positive development of the whole recycling industry.

Third, from the numerical experiments, we can find that there is a negative correlation between the costs and CO_2_ emissions. That is, the reduction in CO_2_ emissions from the DRSC system will negative affect the profitability due to the increased system operation cost. Therefore, in reality, managers should identify the difference between the costs and CO_2_ emissions based on the decision-making with a trade-off. Moreover, the findings of this study can also assist companies in their decision-making. Companies can use the model and solution method proposed to optimally manage their costs and CO_2_ emissions throughout their DRSC system.

Although this paper contributes to the literature on the DRSC with consumer preference, there also exist some limitations. Hence, we will provide some future research. First, this study addresses the dual-objective model with economic profits and environmental sustainability. Furthermore, the social aspects, such as corporate social responsibility, should be also included and discussed in the model formulation due to their high influence on the DRSC operations in a sustainable way. Second, as an exact method, the ε-constraint method is used to solve the stochastic optimization problem; there are many other exact methods and evolutionary algorithms for solving such multi-objective models. Third, the future work can deal with more complex models in strategic network design which integrate uncertainty, multi-product, economics of scale, etc. All these aspects are closer to reality and can provide a more comprehensive analysis for decision-making.

## Figures and Tables

**Figure 1 ijerph-18-04760-f001:**
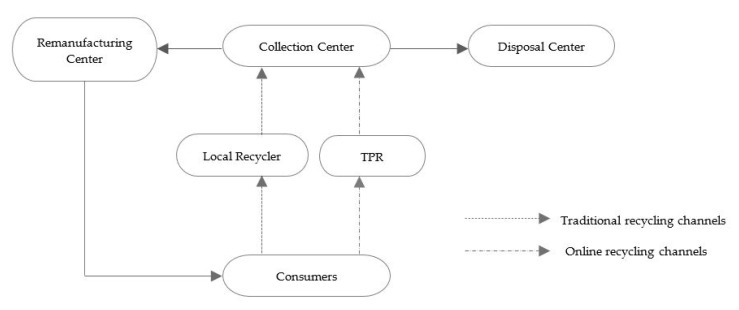
Depiction of the DRSC network.

**Figure 2 ijerph-18-04760-f002:**
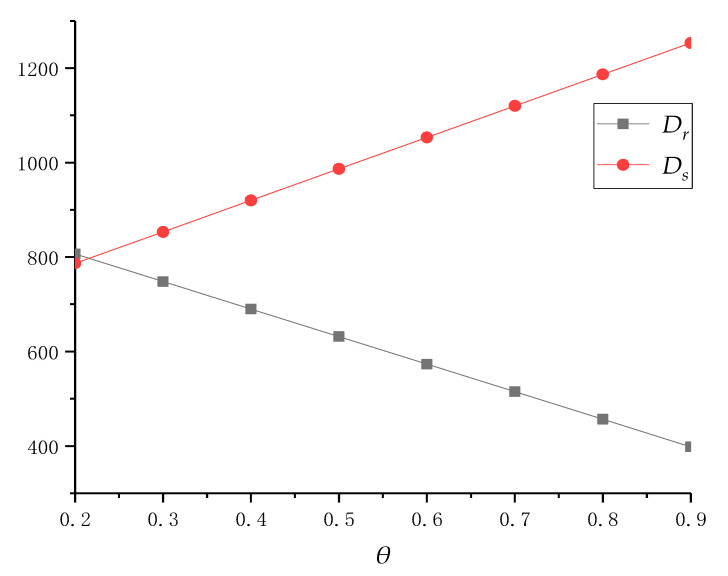
The collective quantity changes in dual channels with different *Θ* when *μ* = 4.

**Figure 3 ijerph-18-04760-f003:**
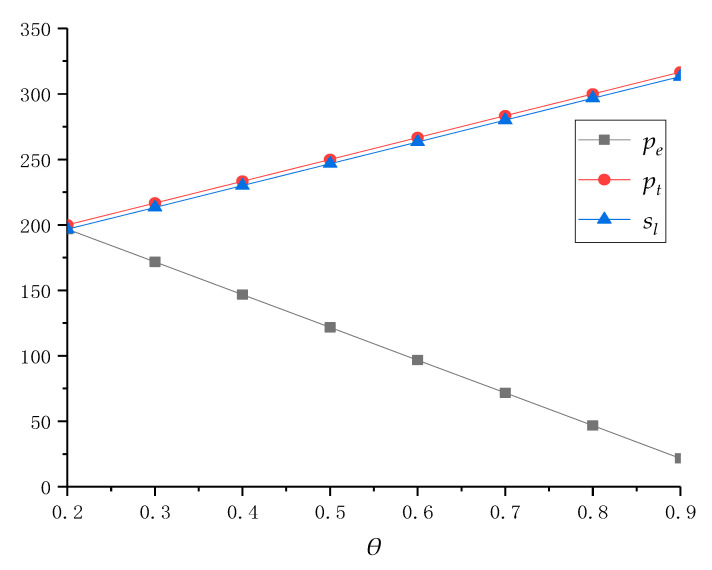
Price and service changes with different *θ* when *μ* = 4.

**Figure 4 ijerph-18-04760-f004:**
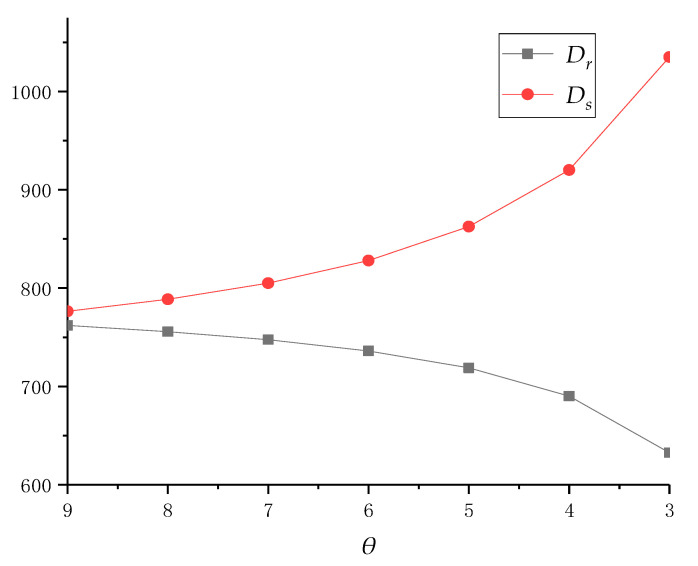
The collective quantity changes in dual channels with different *μ* when *θ* = 0.4.

**Figure 5 ijerph-18-04760-f005:**
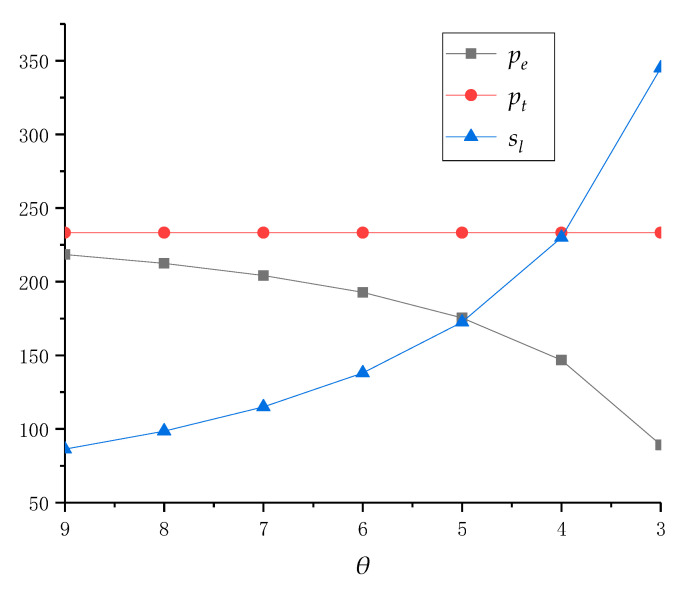
Price and service changes with different *μ* when *Θ* = 0.4.

**Figure 6 ijerph-18-04760-f006:**
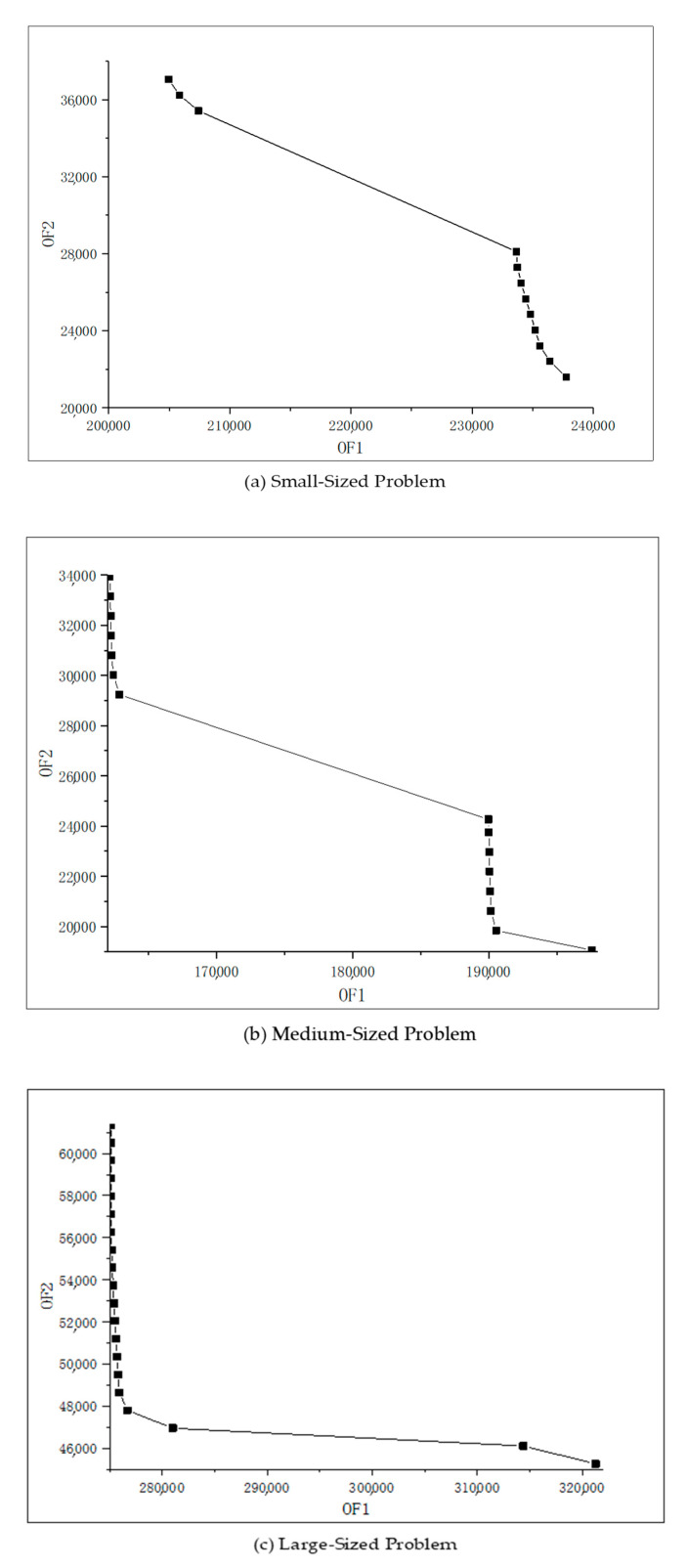
Pareto frontier for three sets of problems: (**a**) small-sized problem; (**b**) medium-sized problem; (**c**) large-sized problem.

**Figure 7 ijerph-18-04760-f007:**
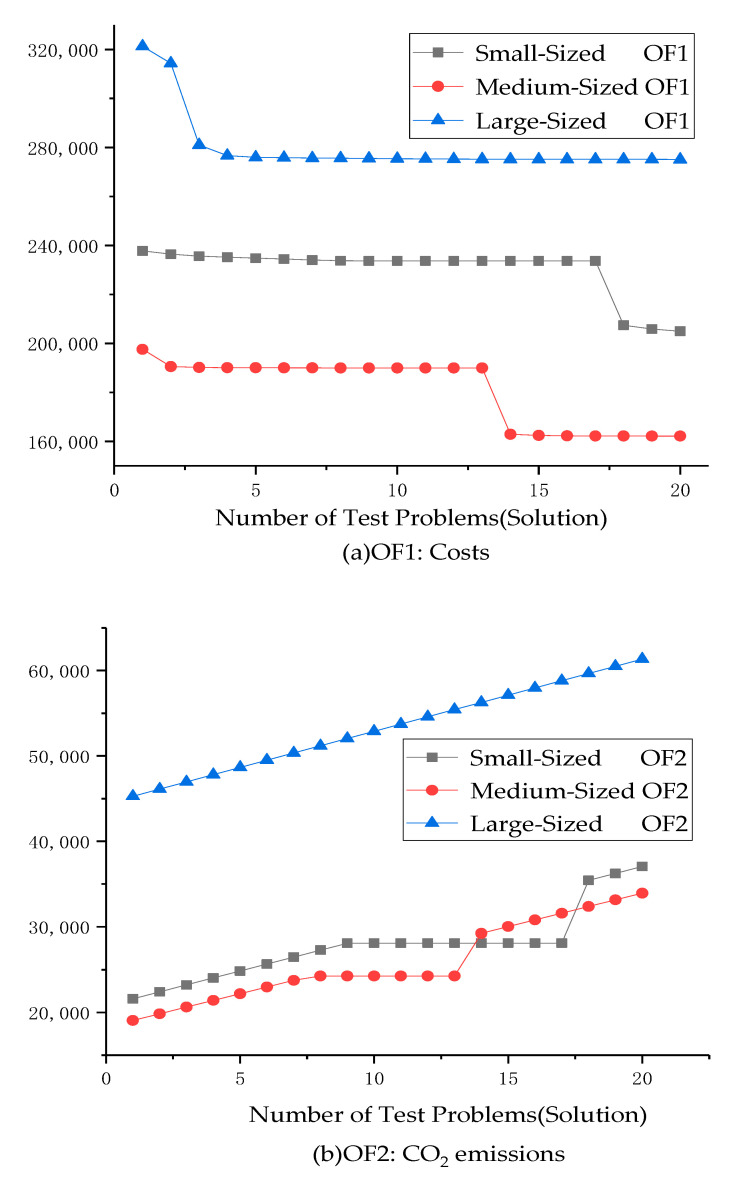
Objective values for three sizes of test problems: (**a**) OF1, (**b**) OF2.

**Table 1 ijerph-18-04760-t001:** Notations.

**Indices**	
*i*	consumers, *i* = 1, 2,…, *I*
*j*	candidates for locations of TPR in online recycling channels, *j* = 1, 2,…, *J*
*k*	candidates for locations of collection centers, *k* = 1, 2,…, *K*
*l*	candidates for locations of remanufacturing centers, *l* = 1, 2,…, *L*
*m*	candidates for locations of disposal centers, *m* = 1, 2…, *M*
*s*	scenarios, *s* = 1, 2,…, *S*
**Parameters**	
*p_s_*	probability of scenario *s*
*β* _1_	recovery rate of remanufacturing center
*β* _2_	remanufacturing rate of remanufacturing center
*FT_j_, FC_k_, FR_l_, FD_m_*	fixed establishing cost of different facilities
*PT_ij_,PC_ik_,PL_jk_,PR_kl_,PD_km_,PU_li_*	unit processing and transporting costs among different facilities
*DT_ij_,DC_jk_,DL_ik_,DR_kl_,DD_km_,DU_li_*	linear distance among different facilities
*ET_j_,EC_k_,ER_l_,ED_m_*	amount of CO_2_ emissions of establishing respective facilities
*HT_j_,HC_k_,HR_l_,HD_m_*	amount of CO_2_ emissions for handling unit e-waste among facilities
*CT_j_,CC_k_,CR_l_,CD_m_*	capacity level for different facilities
*t*	unit CO_2_ emission of shipping one truck-load per kilometer
*ρ*	vehicle capacity occupied by unit of e-waste
**Decision Variables**	
*D_r_, D_s_*	collective quantity of traditional and online recycling channels
*w_j_,*	binary variable which equals ‘1’ if TPR *j* is open, and ‘0’ otherwise
*x_k_*	binary variable which equals ‘1’ if the collection center *k* is open, and ‘0’ otherwise
*y_l_*	binary variable which equals ‘1’ if the remanufacturing center *l* is open, and ‘0’ otherwise
*z_m_*	binary variable which equals ‘1’ if the disposal center *m* is open, and ‘0’ otherwise
*a^s^_ij_, β^s^_jk_, γ^s^_ik_, δ^s^_kl_, η^s^_km_, θ^s^_li_*	amount of e-waste transported among different facilities in scenario *s*

**Table 2 ijerph-18-04760-t002:** The effect of *Θ* on decision and profit.

*Θ*	*p_e_*	*p_t_*	*sl*	*D_r_*	*D_s_*	Π
0.2	196.7	200.0	196.7	806.7	786.7	1,192,070
0.3	171.7	216.7	213.3	748.3	853.3	1,193,480
0.4	146.7	233.3	230.0	690.0	920.0	1,199,070
0.5	121.7	250.0	246.7	631.7	986.7	1,208,820
0.6	96.7	266.7	263.3	573.3	1053.3	1,222,730
0.7	71.7	283.3	280.0	515.0	1120.0	1,240,820
0.8	46.7	300.0	296.7	456.7	1186.7	1,263,070
0.9	21.7	316.7	313.3	398.3	1253.3	1,289,480

**Table 3 ijerph-18-04760-t003:** The effect of *μ* on decision and profit under the centralized mode.

*μ*	*p_e_*	*p_t_*	*sl*	*D_r_*	*D_s_*	Π
9	218.5	233.3	86.3	761.9	776.3	1,149,470
8	212.4	233.3	98.6	755.7	788.6	1,153,720
7	204.2	233.3	115.0	747.5	805.0	1,159,390
6	192.7	233.3	138.0	736.0	828.0	1,167,330
5	175.4	233.3	172.5	718.8	862.5	1,179,230
4	146.7	233.3	230.0	690.0	920.0	1,199,070
3	89.2	233.3	345.0	632.5	1035.0	1,238,740

**Table 4 ijerph-18-04760-t004:** Dimensions of test problems.

Index	Small-Size	Medium-Size	Large-Size
*I*	2	5	10
*J*	2	4	5
*K*	2	3	4
*L*	2	2	2
*M*	1	2	3

**Table 5 ijerph-18-04760-t005:** The main input parameters.

Parameters	Setting
*β* _1_	0.65
*β* _2_	0.5
*t*	0.3 g/ton.km
*ρ*	0.8 ton
*FT_j_*	*FT_j_* = 26,250, 27,500, 33,500, 31,260 and 29,850 RMB for *j* = 1, 2, 3, 4, 5, respectively
*FC_k_*	*FC_k_* = 26,250, 27,500, 33,500 and 31,260 RMB for *k* = 1, 2, 3, 4, respectively
*FR_l_*	*FR_l_ =* 35,000, 28,750 RMB for *l* = 1, 2, respectively
*FD_m_*	*FD_m_ =* 35,000, 28,750 and 30,000 for *m* = 1, 2, 3, respectively
*CT_j_*	*CT_j_* = 1500, 800, 800, 600 and 400, for *j* = 1, 2, 3, 4, 5, respectively
*CC_k_*	*CC_K_ =* 3000, 2000, 1000 and 800 for *k* = 1, 2, 3, respectively
*CR_l_*	*CR_l_ =* 2000, 1000 for *l* = 1, 2, respectively
*CD_m_*	*CD_m_ =* 1000, 500 and 300 for *m* = 1, 2, respectively
*ET_j_*	*ET_j_* = 630, 854, 788, 565 and 1012 g for *j* = 1, 2, 3, 4, 5, respectively
*EC_k_*	*EC_k_* = 1230, 1527, 1328, 1549 g for *k* = 1, 2, 3, 4, respectively
*ER_l_*	*ER_l_* = 2789, 3252 g for *l* = 1, 2, respectively
*ED_m_*	*ED_m_* = 985, 1024 and 892 g for *m* = 1, 2, 3, respectively
*HT_j_*	*HT_j_* = 1.2, 0.8, 1.5, 0.7 and 0.8 g for *j* = 1, 2, 3, 4, 5, respectively
*HC_k_*	*HC_K_ =* 0.9, 1.25, 1.2 and 1.3 g for *k* = 1, 2, 3, respectively
*HR_l_*	*HR_l_ =* 1.75, 1.58 g for *l* = 1, 2, respectively
*HD_m_*	*HD_m_ =* 1.45, 1.23 and 1.67 g for *m* = 1, 2, 3, respectively

**Table 6 ijerph-18-04760-t006:** Linear distance of *DT_ij._*

	*i* = 1	*i* = 2	*i* = 3	*i* = 4	*i* = 5	*i* = 6	*i* = 7	*i* = 8	*i* = 9	*i* = 10
*j* = 1	58.1	41.5	5.7	11	35.6	13.5	10.8	16.7	26.6	15.9
*j* = 2	51.4	17	8.4	5.8	33.6	8.7	4.8	22	17.4	27.5
*j* = 3	60.5	18.8	13.9	7.4	34.1	6.4	8.4	6.5	15.2	19.6
*j* = 4	27.9	20.2	10.8	9.5	39.5	6.5	7	7.9	5.1	16.8
*j* = 5	38.6	15.8	17.7	17.5	37.2	7.1	13	15.5	7.7	24.4

**Table 7 ijerph-18-04760-t007:** Linear distance of *DC_jk._*

	*j* = 1	*j* = 2	*j* = 3	*j* = 4	*j* = 5
*k* = 1	28.9	25.3	40.6	14	34.1
*k* = 2	8.4	11.4	21.2	15.7	9.8
*k* = 3	12.2	8.6	20.3	7.8	44.2
*k* = 4	37.4	15.9	26.3	19.5	10.3

**Table 8 ijerph-18-04760-t008:** Linear distance of *DL_ik._*

	*i* = 1	*i* = 2	*i* = 3	*i* = 4	*i* = 5	*i* = 6	*i* = 7	*i* = 8	*i* = 9	*i* = 10
*k* = 1	47.2	38.2	28.6	19.3	15.5	15.2	23.4	28.5	34.1	40.6
*k* = 2	34.1	24.3	34.8	40.7	40.9	25.8	20.4	23.6	8.4	14
*k* = 3	24.9	26.4	16.3	21.2	44.2	29	34.5	16.2	9.6	49.9
*k* = 4	35.4	27	26.3	27.1	24.4	19.3	14.1	10.1	5	11.4

**Table 9 ijerph-18-04760-t009:** Linear distance of *DR_kl._*

	*k* = 1	*k* = 2	*k* = 3	*k* = 4
*l* = 1	229	253	343	388
*l* = 2	59.8	87.1	47.8	26.7

**Table 10 ijerph-18-04760-t010:** Linear distance of *DD_km._*

	*k* = 1	*k* = 2	*k* = 3	*k* = 4
*m* = 1	78.6	55.7	53.6	46.2
*m* = 2	56.3	100.3	47.8	59.8
*m* = 3	87	120.1	71.3	93.6

**Table 11 ijerph-18-04760-t011:** Linear distance of *DU_li._*

	*l* = 1	*l* = 2
*i* = 1	97	58
*i* = 2	77	38
*i* = 3	87	93
*i* = 4	99	62
*i* = 5	61	79
*i* = 6	58	35
*i* = 7	93	45
*i* = 8	65	58
*i* = 9	81	62
*i* = 10	66	70

**Table 12 ijerph-18-04760-t012:** Objective function values for the small-sized test problem.

Solution	*OF* _1_	*OF* _2_	Solution	*OF* _1_	*OF* _2_
1	237,778.002	21,593.869	11	233,666.023	28,111.686
2	236,427.319	22,407.456	12	233,666.023	28,111.686
3	235,604.077	23,221.043	13	233,666.023	28,111.686
4	235,216.685	24,034.630	14	233,666.023	28,111.686
5	234,829.293	24,848.216	15	233,666.023	28,111.686
6	234,441.902	25,661.803	16	233,666.023	28,111.686
7	234,054.510	26,475.390	17	233,666.023	28,111.686
8	233,744.785	27,288.976	18	207,419.275	35,424.844
9	233,666.896	28,102.563	19	205,853.141	36,238.430
10	233,666.023	28,111.686	20	204,950.853	37,052.017

**Table 13 ijerph-18-04760-t013:** Objective function values for the medium-sized test problem.

Solution	*OF* _1_	*OF* _2_	Solution	*OF* _1_	*OF* _2_
1	197,570.758	19,060.571	11	189,990.757	24,266.359
2	190,543.478	19,843.403	12	189,990.757	24,266.359
3	190,161.481	20,626.235	13	189,990.757	24,266.359
4	190,104.172	21,409.067	14	162,874.151	29,237.389
5	190,073.099	22,191.899	15	162,417.045	30,020.221
6	190,042.026	22,974.731	16	162,277.535	30,803.053
7	190,010.953	23,757.564	17	162,246.462	31,585.886
8	189,990.757	24,266.359	18	162,215.389	32,368.718
9	189,990.757	24,266.359	19	162,184.316	33,151.550
10	189,990.757	24,266.359	20	162,153.243	33,934.382

**Table 14 ijerph-18-04760-t014:** Objective function values for the large-sized test problem.

Solution	*OF* _1_	*OF* _2_	Solution	*OF* _1_	*OF* _2_
1	321,271.269	45,273.026	11	275,329.764	53,736.241
2	314,385.511	46,119.348	12	275,250.502	54,582.563
3	281,020.182	46,965.669	13	275,186.289	55,428.884
4	276,680.972	47,811.991	14	275,144.230	56,275.206
5	275,942.540	48,658.312	15	275,138.076	57,121.527
6	275,800.915	49,504.634	16	275,132.094	57,967.849
7	275,674.417	50,350.955	17	275,127.376	58,814.170
8	275,584.607	51,197.277	18	275,123.362	59,660.492
9	275,499.659	52,043.598	19	275,119.347	60,506.813
10	275,414.712	52,889.920	20	275,116.202	61,353.135

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
