# Peer review of "A Multi-Echelon Network Design in a Dual-Channel Reverse Supply Chain Considering Consumer Preference"

_ijerph, 2021, doi:10.3390/ijerph18094760_

Round 1

Reviewer 1 Report

I congratulate the authors on the manuscript. I only recommend rechecking the language and following the editor's instructions for possible publication.

Author Response

Thank you very much for your comments. In this version, we carefully examined the grammar problems in paper, and also invited English-speaking colleagues to help us re-check the full text. More details of the paper revision are as follows:

Table 1 more details of the paper revision

Line number

Before

after

Description

L93

…to focus on…

…focus on…

delete “to”

L186

…the recycling rate…

… the recovery rate…

“recovery rate” instead of “recycling rate”

…the recycling revenue sharing to…

… the revenue sharing ratio…

delete “recycling”

L253

…we only considered…

… we only consider…

“consider” instead of “considered”

L260

…and cooperated…

… and cooperate…

“cooperate” instead of “cooperated”

L272

…camera…

…cameras…

“cameras” instead of “camera”

L316

…the e-waste…

…the e-waste products…

add “products”

L327

…operating cost…, which consists…

…operating costs…, which consist

“operating costs” instead of “operating cost”, “which consist” instead of “which consists”

L384

…bi-objective model…

… dual-objective model…

“dual-objective” instead of “bi-objective”

L405

…c is the logistics cost of recycling in online channels and cs is the coefficient of service cost…

…c and cs are the logistics cost and service cost of recycling in online channels, respectively…

re-write

L407

cs can be described as…

cs can be described as …where μ represents the coefficient of service cost of online channels…

add “where μ represents the coefficient of service cost of online channels”

L535

…collection center can…

…collection centers can…

“collection centers” instead of “collection center”

L544

…needs…

…need…

“need” instead of “needs”

L547

…prices of traditional…

…price of traditional…

“price” instead of “prices”

L548

…online recycling prices…

…online recycling price…

“price” instead of “prices”

L549

…recycling services…

…recycling service…

“service” instead of “services”

L577

… the optimization of does not always exist all went well…

… the optimization of decisions does not always exist all went well…

add “decisions”

L578

…reduce it from 4 to 3, but it is difficult to reduce it from 9 to 8…

… reduce it from 9 to 8, but it is difficult to reduce it from 4 to 3…

re-write

L592

… in table 6 to 11…

… in table 6 to 11…(Unit: km)…

add “(Unit: km)”

L615

…that the solution…

… that the solutions…

“solutions” instead of “solution”

L616

…bi-objective…

…dual-objective…

“dual-objective” instead of “bi-objective”

L690

…(as shown in Fig.6)…

…(as shown in Figure. 6)

“Figure.” Instead of “Fig.”

L710

…between the cost…

…between the costs…

“costs” instead of “cost”

L734

… the Hessian matrix…Has..

… the Hessian matrix of…as...

“as” instead of “Has”

L744

……

……

“” instead of “”

Reviewer 2 Report

Thank you for the revised version,

Author Response

Thank you very much for your comments. In this version, we carefully examined the grammar problems in paper, and also invited English-speaking colleagues to help us re-check the full text. More details of the paper revision are as follows:

Table 1 more details of the paper revision

Line number

Before

after

Description

L93

…to focus on…

…focus on…

delete “to”

L186

…the recycling rate…

… the recovery rate…

“recovery rate” instead of “recycling rate”

…the recycling revenue sharing to…

… the revenue sharing ratio…

delete “recycling”

L253

…we only considered…

… we only consider…

“consider” instead of “considered”

L260

…and cooperated…

… and cooperate…

“cooperate” instead of “cooperated”

L272

…camera…

…cameras…

“cameras” instead of “camera”

L316

…the e-waste…

…the e-waste products…

add “products”

L327

…operating cost…, which consists…

…operating costs…, which consist

“operating costs” instead of “operating cost”, “which consist” instead of “which consists”

L384

…bi-objective model…

… dual-objective model…

“dual-objective” instead of “bi-objective”

L405

…c is the logistics cost of recycling in online channels and cs is the coefficient of service cost…

…c and cs are the logistics cost and service cost of recycling in online channels, respectively…

re-write

L407

cs can be described as…

cs can be described as …where μ represents the coefficient of service cost of online channels…

add “where μ represents the coefficient of service cost of online channels”

L535

…collection center can…

…collection centers can…

“collection centers” instead of “collection center”

L544

…needs…

…need…

“need” instead of “needs”

L547

…prices of traditional…

…price of traditional…

“price” instead of “prices”

L548

…online recycling prices…

…online recycling price…

“price” instead of “prices”

L549

…recycling services…

…recycling service…

“service” instead of “services”

L577

… the optimization of does not always exist all went well…

… the optimization of decisions does not always exist all went well…

add “decisions”

L578

…reduce it from 4 to 3, but it is difficult to reduce it from 9 to 8…

… reduce it from 9 to 8, but it is difficult to reduce it from 4 to 3…

re-write

L592

… in table 6 to 11…

… in table 6 to 11…(Unit: km)…

add “(Unit: km)”

L615

…that the solution…

… that the solutions…

“solutions” instead of “solution”

L616

…bi-objective…

…dual-objective…

“dual-objective” instead of “bi-objective”

L690

…(as shown in Fig.6)…

…(as shown in Figure. 6)

“Figure.” Instead of “Fig.”

L710

…between the cost…

…between the costs…

“costs” instead of “cost”

L734

… the Hessian matrix…Has..

… the Hessian matrix of…as...

“as” instead of “Has”

L744

……

……

“” instead of “”

This manuscript is a resubmission of an earlier submission. The following is a list of the peer review reports and author responses from that submission.

Round 1

Reviewer 1 Report

Dear authors, I congratulate you for the work developed, I consider it an interesting and new topic.

In order to strengthen the work, I issue the following recommendations:

Literature review
1) Clearly define the theories that support or support the study, I understand that the work addresses the issue of the reverse supply chain. However, this subject must have a theoretical support such as the theory of sustainability, the theory of Stakeholders, etc. In addition, within the variables of the study they consider the internet and the consumer. Therefore, I have a doubt why not base the study from the point of view of the technological adaptation model (TAM), or from the point of view of psychological theory (SIT-Social Identity Theory). These are some ideas that could be incorporated into your work.

2) In the session of assumptions, it is important to justify not only from the empirical point of view but also from the theoretical point of view, to carry out these contrasts in the two aspects in the session of conclusions of the work.

Methodology

1) It is not clear the study subjects and how the analysis was carried out on these subjects. Where did the data come from? It is important to know more precisely the characteristics of the study subjects and the method of data collection.

Conclusions

1) Clarify with greater precision the differences or similarities of their findings with the previous results existing in the literature from the empirical and theoretical point of view.

2) Clarify with greater precision the theoretical and empirical contribution generated by this study. (Researchers, academics, universities, companies, managers, society, consumers, etc).

Author Response

Thank you very much for your very helpful comments. We have made all necessary changes in the revised version of this paper to address all the comments. Below is our response to the comments. For the sake of readability, each original comment (in italic) is inserted before its response.

Q1: Clearly define the theories that support or support the study, I understand that the work addresses the issue of the reverse supply chain. However, this subject must have a theoretical support such as the theory of sustainability, the theory of Stakeholders, etc. In addition, within the variables of the study they consider the internet and the consumer. Therefore, I have a doubt why not base the study from the point of view of the technological adaptation model (TAM), or from the point of view of psychological theory (SIT-Social Identity Theory). These are some ideas that could be incorporated into your work.

Answer: In order to enhance the theoretical base of this paper, we have added the relevant literature on sustainability theory and stakeholder theory in lines 131-139 and 153-165, respectively. Moreover, we believe that the theories of TAM and SIT you mentioned have brought new ideas to our work, which will be incorporated into our work in the future and can be regarded as a new research innovation.

Q2:In the session of assumptions, it is important to justify not only from the empirical point of view but also from the theoretical point of view, to carry out these contrasts in the two aspects in the session of conclusions of the work.

Answer: According to your comment, we have revised this part in the paper.

Methodology

Q3: It is not clear the study subjects and how the analysis was carried out on these subjects. Where did the data come from? It is important to know more precisely the characteristics of the study subjects and the method of data collection.

Answer: We have elaborated on the study subject and how the analysis was carried out on these subjects in lines 221-240. And for the setting of test data, we have made a more specific description in Section 5.2.  

Conclusions:

Q4: Clarify with greater precision the differences or similarities of their findings with the previous results existing in the literature from the empirical and theoretical point of view.

Answer: For this problem, we have revised the main contributions to highlight the research goal and the main findings of this work, as well as the difference from previous studies.

Q5: Clarify with greater precision the theoretical and empirical contribution generated by this study. (Researchers, academics, universities, companies, managers, society, consumers, etc).

Answer: On the basis of existing conclusions in this paper, we have added a discussion in Section 5.3 to strengthen the motivation and contribution.

Finally, we must express our thanks to you again. All the constructive comments you provided are very helpful to improve the manuscript. It is also believed that after the revision, this paper can fulfil requirements of you and the editor, and can also reach a high level for publication. We wish you all the best in your future work.

Reviewer 2 Report

The article A Multi-Echelon Network Design in Dual-Channel Reverse 2 Supply Chain Considering Consumer Preference is an interesting one. The article covers a wide set of disciplines. The following comments will help improve the manuscript.

  1. The introduction section is reasonable, written well, but very narrowly focussed, therefore the authors must check to make it more relevant to the journal.
  2. The research question(s) are not clear, if the authors can add research objectives following research contributions then it will increase readability.
  3. Research assumptions need more theoretical base. Author(s) should link the assumptions to existing literature.
  4. It would be nice to see some kind of conceptual framework for this research.
  5. I would suggest refining your research methodology. At the moment, rather limited discussion on research philosophy.
  6. Critically analyse different theoretical approaches to the research problem
  7. Justify the solution method selected in terms of the research objectives
  8. Strengthen the motivation and discussion parts substantially.
  9. Demonstrate adequately that your solution findings have been logically derived and that conclusions, solutions/recommendations are fully supported by evidence
  10. While Reverse Supply Chain is still a budding topic, still one can find many research papers in other reputed journals as well. I'd suggest referring those articles as well as very few literatures has been cited in the current manuscript.
  • White, G., Svetlana. C., Subramanian, N., and Dwivedi. A. 2018. Soft side of knowledge transfer partnership between universities and small to medium enterprises: Exploratory study to understand process improvement. Production Planning & Control: The Management of Operations
  • Liang, T., Mohan, S. and Subramanian, N. 2018. Environmental improvement initiatives in the coal mining industry: Maximisation of the triple bottom line. Production Planning & Control: The Management of Operations
  • Upadhyay, A. 2020. Antecedents of green supply chain practices in developing economies. Management of Environmental Quality: An International Journal.
  • Jaeger, B., Menebo, M. 2021. Identification of environmental supply chain bottlenecks: A case study of the Ethiopian Healthcare Supply Chain. Management of Environmental Quality: An International Journal.
  • Kumar, N., Brint, A., Shi, E, Ximing, P. 2018. Integrating sustainable supply chain practices with operational performance: An exploratory study of Chinese SMEs. Production Planning & Control: The Management of Operations
  • Jaeger, B., Upadhyay, A. 2019. Understanding Barriers of Circular Economy: Cases from the Manufacturing Industry. Journal of Enterprise Information Management
  • Upadhyay, A., Mukhuty, S., Kumar, V. and Kazancoglu, Y. 2021. Blockchain Technology and the Circular Economy: Implications for Sustainability and Social Responsibility. Journal of Cleaner Production
  1. The authors must check all references carefully, there are some references are cited in the text and vice versa.

Author Response

Thank you very much for your very helpful comments. We have made all necessary changes in the revised version of this paper to address all the comments. Below is our response to the comments. For the sake of readability, each original comment (in italic) is inserted before its response.

The article A Multi-Echelon Network Design in Dual-Channel Reverse 2 Supply Chain Considering Consumer Preference is an interesting one. The article covers a wide set of disciplines. The following comments will help improve the manuscript.

Q1: The introduction section is reasonable, written well, but very narrowly focused, therefore the authors must check to make it more relevant to the journal.

Answer: In order to make this paper more relevant to the journal, we have added some descriptions of environment, green and sustainability in lines 33-38 and 51-59. At the same time, when describing the main contributions of this work, we also emphasized the goal of environmental sustainability by reducing CO2emissions.  

Q2: The research question(s) are not clear, if the authors can add research objectives following research contributions then it will increase readability.

Answer: We have revised the main contributions of this work to highlight the research goal and the main findings of this work, as well as the difference from previous studies.

Objectives

Q3: Research assumptions need more theoretical base. Author(s) should link the assumptions to existing literature.

Answer: In order to enhance the theoretical base of assumptions, we have added the relevant literature and revised this part in the paper.

Q4: I would suggest refining your research methodology. At the moment, rather limited discussion on research philosophy.

Answer: We have added a detailed description of the research subject and methodology in the Introduction. According to your valuable advice, we will attempt to refine the research methodology in the future work.

Q5: Critically analyze different theoretical approaches to the research problem.

Answer: In this paper, the aim of developing a dual-objective optimization is to optimize two conflicting objective functions simultaneously. It should be mentioned that there are two classes of methods to optimize the presented bi-objective model: the first one is using meta-heuristics or evolutionary methods which can yield acceptable solutions. However, the quality of the solutions and their optimality are not known. The other type of methods is the exact or heuristic methods used to obtain the Pareto solutions. In this study, the latter method, an epsilon-constraint approach, is applied to solve the model. In our future work, we will attempt to use other method to solve such problem, i.e., Lagrangian-relaxation method, sampling average approximation method and genetic algorithm, etc.

Q6: Justify the solution method selected in terms of the research objectives.

Answer:We have justify the solution method selected in Section 5.

Q7: Strengthen the motivation and discussion parts substantially.

Answer: On the basis of existing conclusions in this paper, we have added a discussion in Section 5.3 to strengthen the motivation and discussion part.

Q8: Demonstrate adequately that your solution findings have been logically derived and that conclusions, solutions/recommendations are fully supported by evidence.

Answer: for this problem, we have discussed in detail from the study subjects, research background, objectives, and methodologies. We also explained the data generation method and how to use it to identify the solution method. And in the Section 5.2 and 6, we put forward a discussion based on the results of the example analysis. All these can be used as evidence to show the main finding of this work.

Q9: While Reverse Supply Chain is still a budding topic, still one can find many research papers in other reputed journals as well. I'd suggest referring those articles as well as very few literatures has been cited in the current manuscript.

  • White, G., Svetlana. C., Subramanian, N., and Dwivedi. A. 2018. Soft side of knowledge transfer partnership between universities and small to medium enterprises: Exploratory study to understand process improvement. Production Planning & Control: The Management of Operations
  • Liang, T., Mohan, S. and Subramanian, N. 2018. Environmental improvement initiatives in the coal mining industry: Maximisation of the triple bottom line. Production Planning & Control: The Management of Operations
  • Upadhyay, A. 2020. Antecedents of green supply chain practices in developing economies. Management of Environmental Quality: An International Journal.
  • Jaeger, B., Menebo, M. 2021. Identification of environmental supply chain bottlenecks: A case study of the Ethiopian Healthcare Supply Chain. Management of Environmental Quality: An International Journal.
  • Kumar, N., Brint, A., Shi, E, Ximing, P. 2018. Integrating sustainable supply chain practices with operational performance: An exploratory study of Chinese SMEs. Production Planning & Control: The Management of Operations
  • Jaeger, B., Upadhyay, A. 2019. Understanding Barriers of Circular Economy: Cases from the Manufacturing Industry. Journal of Enterprise Information Management
  • Upadhyay, A., Mukhuty, S., Kumar, V. and Kazancoglu, Y. 2021. Blockchain Technology and the Circular Economy: Implications for Sustainability and Social Responsibility. Journal of Cleaner Production

Answer: We have cited the above articles to enhance the theoretical basis of this paper.

Q10: The authors must check all references carefully, there are some references are cited in the text and vice versa.

Answer:thank you very much for reminding us this problem, we have checked all the references carefully.

Finally, we must express our thanks to you again. All the constructive comments you provided are very helpful to improve the manuscript. It is also believed that after the revision, this paper can fulfil requirements of you and the editor, and can also reach a high level for publication. We wish you all the best in your future work.
